# On the Fabrication of High-Performance Additively Manufactured Copper Winding Using Laser Powder Bed Fusion

**DOI:** 10.3390/ma16134694

**Published:** 2023-06-29

**Authors:** Mohamed Abdelhafiz, Ali Emadi, Mohamed A. Elbestawi

**Affiliations:** 1Additive Manufacturing Group (AMG), Department of Mechanical Engineering, McMaster University, 1280 Main Street West, Hamilton, ON L8S 4L7, Canada; elbestaw@mcmaster.ca; 2Department of Electrical & Computer Engineering, McMaster University, 1280 Main Street West, Hamilton, ON L8S 4K1, Canada; emadi@mcmaster.ca

**Keywords:** copper, laser powder bed fusion, filling factor, DC and AC resistance, dimensional accuracy

## Abstract

Due to its exceptional electrical and thermal conductivity, pure copper is frequently employed in industry as the base metal for thermal management and electromagnetic applications. The growing need for complicated and efficient motor designs has recently accelerated the development of copper additive manufacturing (AM). The present work aims to improve the power density of the copper laser powder bed fusion (Cu-LPBF) coil by increasing the slot-filling factor (SFF) and the electrical conductivity. Firstly, the dimensional limitation of Cu-LPBF fabricated parts was identified. Sample contouring and adjusting beam offset associated with optimum scan track morphology upgraded the minimum feature spacing to 80 μm. Accordingly, the printed winding’s slot-filling factor increased to 79% for square wire and 63% for round wire. A maximum electrical conductivity of 87% (IACS) was achieved by heat treatment (HT). The electrical impedance of full-size Cu-LPBF coils, newly reported in this study, was measured and compared with solid wire. It can reflect the performance of Cu-LPBF coils (power factor) in high-frequency applications. Furthermore, surface quality benefited from either sample contouring and HT, where the side surface roughness was lowered by 45% and an additional reduction of 25% after HT.

## 1. Introduction

Laser powder bed fusion (LPBF) is a type of AM technique in which a laser is used to selectively melt and fuse metallic powders layer by layer to produce complex-shaped components. The development of AM techniques allows for optimizing electrical machine performance by reducing design constraints existing in traditional manufacturing methods. Copper winding in electrical motors can have intricate shapes and geometries that improve electromagnetic properties and reduce losses [1,2]. However, copper powder has a unique processing behavior due to its high thermal conductivity and reflectivity in the infrared region, where most fiber lasers operate, making it challenging to process using LPBF. Nevertheless, recent research has shown that near-full-density copper parts can be produced using conventional infrared fiber lasers with a laser power of 500 W within a volumetric energy density window of 230–310 J/mm^3^ [3].

AM techniques may not achieve the same physical properties of fabricated parts as traditional manufacturing methods. The less density of copper parts processed by low and medium laser power machines is the main reason behind the low electrical conductivity of those parts. Many researchers have discussed possible ways to overcome this challenge, such as utilizing high laser power [3] or pulsed laser [4] and powder surface modification [5]. Dimensional accuracy (DA) and surface roughness are other drawbacks that can arise in AM parts, particularly when printing a challenging material such as copper [6,7].

Lack of DA is a critical aspect of LPBF products caused by various factors. One prominent reason is the thermal behavior of the material during the build process. During the melting and solidification cycle, thermal stresses and residual heat can cause part size distortion and lead to deviations from the intended dimensions [8]. Additionally, unmelted particles adhered to the solidified part and improper beam offset can contribute to dimensional errors. The beam offset compensates for the heat-affected zone around the laser spot diameter. It is therefore recommended to utilize an optimal beam offset that is half the radius of the laser beam’s curing zone and should be directed inward [9]. Measuring the DA of AM parts starts with designing the benchmark part that encompasses all critical features of interest. Various measuring techniques can be employed to assess DA in additive manufacturing. These techniques include the use of coordinate measuring machines, X-ray computed tomography systems, and optical scanners [10]. Microscope images in conjunction with image processing software offer an alternative noncontact measurement approach. High-resolution microscope images of the part can be captured and processed using specialized software such as ImageJ [11,12]. The software employs image analysis algorithms to extract planar dimensional information such as distances, angles, and feature sizes. This method is advantageous for capturing fine details and can provide automated measurement capabilities.

The surface quality of LPBF parts relies on various factors such as laser parameters, powder characteristics, and part surface inclination angle. Most studies that offer enough information on the process parameters see a general pattern in which the roughness and porosity follow a U-shaped curve with respect to the energy density [8,13]. As reported, high energy density, associated with high laser power and slow scanning speeds, can lead to significant melt pool dynamics, potentially resulting in a rougher surface. Therefore, printing Cu using a medium laser power range at low scanning speed does not compensate for the lack of required energy, since it harms both surface quality and part relative density [7]. In addition, the hatching pattern influences the development of surface roughness; for example, surface irregularity accumulates layer by layer due to nonrotating scan vectors [14]. The presence of partially melted particles, whether at the top or side surface, is a form of roughness that usually accompanies powder-based AM parts. Sample contouring is rarely reported in Cu-LPBF, despite being an effective tool for reducing the roughness of part sides [13,15].

The main objective of this work is to improve the power density of the Cu-LPBF coil by enhancing both the SFF and the electrical conductivity. The coil power density is defined by the amount of electrical power per unit cross-sectional area of the conductor. Increasing the SFF is the most powerful method to improve the power density of electrical motors [16]. In addition, heat generated during operation could be effectively dissipated by the additional copper in the slot. The SFF can be optimized by reducing the spacing between coil turns. In this regard, the DA and surface quality should be optimized first. While some studies have reported the DA and surface roughness of Cu-LPBF parts [7,17,18,19], none have presented potential in situ solutions for improvement. Besides, the minimum spacing between features has not yet been investigated. The first step aims to establish a reference point by identifying the dimensional limitation of uncontoured and unadjusted beam offset Cu-LPBF test coupon. In this step, DA and surface roughness were also investigated. The morphology and size of single-scan tracks were characterized thereafter to obtain the optimum parameters for sample contouring and beam offset. It was followed by printing coils using the obtained minimum spacing to be used for DC and AC resistance measurements. Finally, these coils were heat-treated to quantify the improvement in electrical conductivity.

## 2. Materials and Methodology

### 2.1. Powder Characteristic

In this study, a nitrogen-atomized pure copper powder (99.9% wt) was processed by LPBF to fabricate test samples. The particle size varies from 7 to 60 μm, where the D10, D50, and D90 are equal to 16, 31, and 51 μm, respectively. Powder morphology shows that Cu particles are primarily spherical and quasi-spherical, as illustrated in Figure 1. High magnification on Cu powder indicates a few small particles (satellites) adhesion to the larger ones. During the atomization process, the coarser particles take relatively longer to be solidified than the fine particles. Therefore, it increases the bonding chance with fine particles due to in-flight collision [20]. The Cu powder’s apparent density of 5.07 g/cm^3^ and hall flow of 11 s/50 g were measured by the powder’s supplier according to ASTM B212 and B213, respectively.

### 2.2. LPBF Parameters and Sample Characterization

The EOS M280 LPBF machine was utilized to process the Cu samples for this study. The nominal laser beam diameter was 100 μm, with a continuous wavelength of 1080 ± 20 nm. The test samples were made using optimum laser power of 370 W, scanning speed of 500 mm/s, hatch spacing of 0.1 mm, and layer thickness of 0.04 mm, as reported in [7].

Since the DA of LPBF fabricated parts is strongly influenced by the base material, powder characteristics, and process parameters being adopted, examining the geometric deviation is an essential step prior to printing a product used in a dimension-sensitive application. In this regard, a custom coupon design, inspired by benchmark artifacts presented in [21], was fabricated to determine the minimum acceptable spacing between the features. Figure 2 shows the front and top views of the CAD design. It comprises trapezoidal protrusion features (TF) and three blind holes with diameters of 0.3, 0.6, and 0.9 mm. The scan track width and feature thicknesses were optically measured with ImageJ software. The threshold value of the scanning electron microscope (SEM) images were manually tuned until the interest area border was well defined. The binary images were discretized to obtain one reading per pixel. The surface roughness of the printed samples was quantitively assessed by the Mitutoyo SJ-410 stylus profilometer [8]. Five measurements were done over a set distance of 4.8 mm in the building direction (Z).

The next step was printing four replicas of flat square and round cross-section coils and cube samples, as shown in Figure 3. All parts were fabricated on a sacrifice printed base of 1 mm anchored to the steel substrate to eliminate the negative impact of diffused impurities. The relative density (RD) of cube samples was measured according to the Archimedes principle and following the steps offered by ASTM B962-14. For comparison, the RD of the coils was measured following the same procedure in order to investigate the size effect on part density. The weight of Cu parts in air and deionized water was measured three times at room temperature using AB204 Mettler balance with 0.1 mg resolution. The standard density of solid Cu used in the calculation of RD is 8.89 g/cm^3^. Surface topography and sample cross section were depicted using TESCAN VP SEM powered by secondary electrons at 20 kV (accelerating voltage). The samples were ground using a sequence of 800, 1200, 2400, and 4000 SiC abrasive papers, followed by a polishing process using 3, 1, and 0.05 μm suspended particles size. The polished samples were etched chemically at 20 °C and atmospheric pressure by a reagent of 50 mL of nitric acid (1.40 pH) and 50 mL of distilled water. The samples were submerged in the solution for 2 s, followed by a water rinse. The microstructure was investigated via an optical microscope Nikon LV100.

HT of Cu-LPBF samples was conducted to upgrade the electrical conductivity by retrieving the typical microstructure of copper. A horizontal tube furnace with an argon inert gas atmosphere was used to heat the samples to 1000 °C, where the temperature ramped up by 10 °C per min. The dwell time was 4 h, followed by furnace-cooling to ambient temperature.

The DC resistance of as-built and (heat-treated) HTed coils was measured by a KEITHLEY 2400 Source Meter from the Tektronix company at room temperature. The Four-Terminal Sensing method was implemented to counteract the impact of contact resistance on the readings (Figure 4a). In addition, 12-gauge (2.05 mm) copper wires with round and square cross sections were procured to use as a reference in this study, noted as “solid” wire. These high-purity Cu wires (99.9%) were manufactured by cold-drawing followed by fully annealed heat treatment. The selection of dead soft wire is due to its high flexibility and best conductivity (approach 100% IACs). The AC resistance (impedance) was measured for all coils using the Impedance Analyzer KEYSIGHT E4990A, which produces frequencies of a range from 20 Hz to 10 MHz (Figure 4b).

## 3. Results and Discussion

### 3.1. Dimensional Accuracy and Minimum Spacing

#### 3.1.1. No Contouring and Unadjusted Beam Offset

In light of the current work’s goal of examining the minimum feature size and DA, It is necessary to establish a baseline using an untuned sample (no contouring and without beam adjustment) for future comparisons. Figure 5 shows a high variation in feature width compared with the nominal size of the CAD file. The width of the isosceles trapezoids varied linearly in CAD design from 0.2 mm to 1 mm, where the gap formed sunken triangles of a 0.8 mm base. The enlargement of width after LPBF was noted along the TF length regardless of the nominal size. The average deviation was +80 μm, with a maximum and minimum value of 200 and 0 μm, respectively. The minimum spacing was measured after image conversion to a binary format which varied from 80 μm up to a maximum of 180 μm, as shown in Figure 6. The minimum spacing between TFs is defined based on gap closing at the first and last contact points, since the TFs were noted to be reopened in some instances, as depicted in Figure 5. The last contact point exhibited less variation of minimum spacing with a standard deviation of 13.6 μm and less average of 93 μm.

#### 3.1.2. Application of the Beam Offset and Contouring

The previous section assures the need to reduce surface roughness and average size distortion of the LPBFed Cu part. Figure 7 shows a schematic of LPBF part size before and after employing contouring and beam offset optimization. The width of scan tracks depends on laser parameters and material properties. Therefore, the beam offset of Cu-LPBF parts made by the high laser power available on the EOS machine will be adequately specified to compensate for the thickness of the single-track width. The employed laser follows an axisymmetric Gaussian profile characterized by the distribution factor indicating the laser power intensity at the beam center and the radial decay of laser power. As a result, the nominal beam diameter is surrounded by a heat-affected zone noted by the effective laser beam diameter, as presented in Figure 7. The distribution factor is seldom published for SLM machines. In addition, the focus diameter for the utilized laser at the exposed area may vary between 100 and 500 μm. Thus, the track width is indeed difficult to predict. Therefore, it will be measured based on the actual printed single-scan track. The outer surface irregularity of uncontoured Cu parts is due to the heterogeneity of heat energy delivered along the part edge, as illustrated in Figure 7a. The laser path’s start and end points also possess different energy densities due to the acceleration and deceleration of the laser scanner [9]. In contrast, the contour track guarantees consistent melting of the part edge.

Assessing the scan line width and morphology

Figure 8a shows the scan tracks of Cu processed by laser power of 370 W and scanning speed from 200 to 800 mm/s. A black background was used instead of the substrate images for better representation and increasing measurement accuracy (Figure 8b). Under SEM, it was observed that reducing the scan speed leads to enlarging the track width due to the rising in linear energy density. For example, the maximum average width was found to be 300 μm at the corresponding speed of 200 mm/s. It is triple the size of the nominal laser beam diameter. The maximum absolute width is 440 μm, promoted by the high energy and the adhesion of Cu particles. Figure 9 summarizes the change in average width relative to scanning speed. Surprisingly, the minimum average width of 170 μm is still more than the beam diameter.

The scanning speed and the corresponding linear energy density also impact the morphology of tracks. As shown in Figure 8b, the scan tracks become more unstable by increasing the scanning speed. The presence of high- and low-pressure areas that result in pinched and humped bead zones, indicated in Figure 8b, implies that the melt pool is disturbed. Due to insufficient melting and lack of wettability, the surface tension of the melt pool during solidification introduces an axisymmetric periodic perturbation as determined by Plateau–Rayleigh stability [22,23]. Further reduction of linear energy density at the highest V of 800 mm/s can lead to fragmentation of the melt line. The width fluctuation is plotted along the investigated length of the deposited tracks starting from reference coordinate (x), as shown in Figure A1. The minimum variance of track width was obtained at V equal to 400 mm/s.

The beam offset and scanning speed assigned for contouring and core process parameters are stated in Table 1 according to the results acquired in this section and the density measurement obtained in [7]. Since the partially melted powder particles adhered to the track border were observed at all scanning speeds, cy the beam offset during core melting is shifted a little further by the magnitude of maximum deviation aiming to reduce any additional powder adhesion. It should be noted that although the scanning speed of 500 mm/s is the optimum value from the part density point of view, it results in a slightly higher track width variation. LPBF is a very complex thermodynamic process. Therefore, it is impossible to judge or predict optimum performance based on a single parameter, but it is an integration of the influence of multiple factors.

Effect of sample contouring on side surface roughness

Surface roughness significantly improved by applying sample contouring. The arithmetic average roughness (Ra) measured by a stylus profilometer showed a remarkable reduction by half. The SEM micrograph of lateral surfaces presents a secondary structure of adhered powder particles due to incomplete fusion (partially melting), as shown in Figure 10. Those particles were observed in both cases, as depicted in the two magnified views of Figure 10. Although the energy density of the contour path is higher than that of the open-core surface, particle adhesion is more intensive in the uncontoured samples. It is attributed to the sizeable irregular surface area created by meltpool spheroidization at the tracks terminal. The formation of balls, as indicated in Figure 10b, is the most significant factor affecting surface quality, i.e., Ra was improved from 9.5 ± 0.7 to 5.3 ± 0.5 μm, corresponding to grade N10 and N9, respectively. This conspicuous behavior can be explained by the lack of wettability of molten metal with adjacent tracks and the solidified layer at track terminals. Particularly, LPBF of pure copper with medium laser power is a metastable process, as evidenced by the narrow process parameter region [7]. Melt pool instability at the track terminal might also result in splashes of molten droplets [24]. These formed balls take irregular shapes with approximately 100 μm in diameter. On the other hand, excellent layers bonding can also be achieved by contouring, as evinced by the detectable deposited layers’ interface in the magnified view of Figure 10a.

The edge morphology of cross-sectioned DA coupons in the XY plane is illustrated in Figure 11 with and without contouring. It reveals a new location of lack of fusion defect, approximately 200 μm away from the edge, aligning with the contour/hatch interface. The overlap between the contour and core hatch is reported to be one of the reasons behind such a defect [24]. According to the track width measurement and the applied beam offset, the overlapping of 70 μm can be assumed to provide successful bonding. It is most likely caused by scan line instability where the melt pool (liquid) collides with the solidified contour. At this instant, the molten metal experienced a different surface force and heat transfer mechanism affecting the wettability and morphology. Adding to that, the high thermal conductivity of copper diminished the opportunity to remelt the solidified contour at the overlap region. A sharp valley defect is observed at the surface of the uncontoured samples. It reduces the mechanical strength by increasing the stress concentration at this point. Therefore, it offers a further benefit for contouring use.

Effect of beam offset adjustment on DA

The minimum spacing between the TF was improved to be around 80 μm after applying sample contouring and the optimum beam offset of 110 μm. Figure 12 shows a consistent minimum spacing compared with the uncontoured sample. It is worth noting that the width variation is size-dependent, as shown in the TF width, versus the length plot in Figure 12. For example, smooth surface and accurate dimension compared with the cad size were found in the thicker region (width above 0.5 mm). On the contrary, the thinner part of all the TFs less than 0.3 mm exhibit a higher size error with randomly distributed unfused powder particles. Despite being contoured, the unfused particles cannot be avoided, which increases the uncertainty of the surface quality and size deviation of the Cu-LPBFed parts.

Since the coil turns may consist of straight and curved parts, this section aims to briefly present the out-of-roundness error for a set of holes printed on the Cu-LPBF coupon, as shown in Figure 13a. The circularity (roundness) of microhole features was calculated using Cox’s method by defining circle equivalent diameter, as in Equation (1), where A is the actual surface area of the object, and P is the perimeter (Figure 13b) [20]. Therefore, the perfect roundness will be at $\phi_{cox}$ equal to 1. The actual polygon area (A) was calculated by summing the triangle areas corresponding to two consecutive points on the circumference $\left(x_{i},y_{i}\right)$, $\left(x_{i+1},y_{i+1}\right)$ and the center $\left(x_{c},y_{c}\right)$, known by gauss’s area or shoelace formula. A different method of evaluating roundness error was also conducted by means of Minimum Zone Circle (MZC), which is the difference between the radius of the smallest circumscribed circle (R_c_) and the radius of the largest inscribed circle (R_i_) [25]. As defined by ASME Y14.5 M-1994, the two concentric circles are assigned in a way that minimizes the radial distance [25]. Table 1 shows the $\phi_{Cox}$ compared with MZC with respect to three hole sizes.
(1)$$\phi_{cox}=4\pi A/P^{2}$$
where $A=0.5\times \sum_{i=1}^{n}\left|\begin{array}{cc} x_{i} & x_{i+1}\\ y_{i} & y_{i+1} \end{array}\right|$.

The calculation of roundness error obtained by $\phi_{cox}$ and the MZC method are provided in Figure 14. Even though contouring of curved edges significantly raises the quality of the surface, out-of-roundness continues to pose an issue. It is worth noting that hole accuracy slightly improves as the size increases according to Cox criteria. It is assumed to be due to the fluctuation of the scan track during the contouring of a small curvature radius. The MZC technique exposes a broad range of radial deviations between the min and max locations, which correspond to random particle adhesion and track perturbation from one deposited layer to another. The maximum size error obtained in this study is 200 μm. Therefore, it needs to be considered when designing the Cu-LPBF coils.

### 3.2. Characterization of As-Built Coil

The LPBF-Cu coils used in this study were designed in accordance with several requirements: (1) fabrication on top of 1 mm of sacrifice printed Cu substrate to avoid any upward fusion of metallic impurities, i.e., Fe and C [7]. (2) the minimum spacing between the coil turns was selected to be 250 μm, which gives more tolerance at the curved region and triple the margin of the straight portion. The goal is to prevent any short contact between coil turns that may be caused by unexpectedly adhered particles, as discussed previously. (3) The coil length should be compatible with the ASTM B193-02 Standard, which recommends the test length to be at least 300 mm [26]. (4) Thinner wire was noted to affect the RD negatively, so a 2 mm wire thickness (square and round) with acceptable properties was used.

#### 3.2.1. RD and SFF of Cu Coils

The measured RD of cube samples, round, and square cross-section wires was 95.7 ± 0.26%, 93.9 ± 0.19%, and 94.3 ± 0.22%, respectively. The RD of purchased solid wire was also measured to be 99.95 ± 0.15%. Despite the fact that all parts were processed with the same laser parameters and measured using the same procedures and methodology, the RD of the as-built coils indicates a slight drop when compared to the cube samples. Increasing the surface-to-volume ratio of LPBFed parts made of high thermal conductivity material such as Cu leads to different densification between core and border regions, as shown in Figure 15. The exterior surface of the LPBF product is always in direct contact with the unfused powder that prompts heat dissipation, particularly when applying a lower preheating temperature. On the other side, heat accumulation in the core causes the peak temperature to increase, which improves the RD of Cu [27].

The SFF is defined as the ratio of the cross-sectional area that the conductor wire occupies within the stator slot to the total amount of empty slot space [28]. Assuming that the coil will be installed in a rectangular stator slot with 200 μm tolerance in all directions, the SFF was calculated to be 79% and 63% for square and round wire, respectively. In comparison with the traditional manufacturing method, the hairpin winding of square wire, developed recently to improve the power density of electric motors by optimizing the wire packing, can provide an SFF of 75% for square wire [29]. The benefit of employing LPBF in winding fabrication is the degree of freedom in designing a wire cross section that optimizes the SFF. Additionally, various insulation types can be used to fill the wires spacing of printed coils, enhancing performance and reliability.

#### 3.2.2. DC and AC Resistance

The electrical conductivity (σ) of as-built round and square winding was calculated by substituting in Equation (2) using the measured DC resistance $(R_{DC}$) and wire geometries. The standard electrical conductivity ($\sigma_{o}$) of pure wrought Cu, according to the International Annealing Standard (IACS), is 5.8 × 10^7^ S/m [26]. The actual cross-sectional areas (A) of all wires used in the calculation are presented in Figure 16a. It is noticeable that the solid square wire has a small round fillet. Therefore, the printed square coil was ground using fine sandpaper to obtain a comparable size. Figure 16b shows the σ of the printed coil compared with the corresponding solid winding (pure wrought). The σ of LPBFed Cu winding was 20% less compared with the standard conductivity of Cu.
(2)$$\sigma \left(IACS\right)=\frac{l/(R_{DC}A)}{\sigma_{o}}$$

When electrons move through a conductor such as copper, they experience collisions with impurities and other imperfections in the crystal lattice. These collisions lead to the electrons’ scattering, increasing their resistance to flow. In the case of solid pure metal, the electrical resistance is governed by the number of disruptions found in the periodic atomic lattice structure [30]. It reduces the free-traveling distance of electrons without disturbance, known as the “mean free path”. The increase in resistivity of the LPBFed part over that of pure solid metal originated from material defects such as porosity, grain boundaries, vacancies in the atomic lattice, and dislocation of crystal structure. The porosity (when present) in Cu-LPBF has the greatest impact on material conductivity [31]. Electrons may collide with the walls of the pores as they pass through the material. These collisions create a zigzagging path for the electrons, which increases the traveling distance, leading to an overall increase in resistance. Additionally, the porosity reduces the effective cross-sectional area. Several researchers developed models that describe the relationship between material conductivity and porosity (ε), as stated in Table 2. Equation (3a) is a straight line drawn to fit the experimental data for Cu processed by powder metallurgy of seven researchers [32]. In a range of ε less than 30%, the relationship can also be approximated by a straight-line equation but with ω=1.123 [33]. The effective thermal conductivity ($K_{eff}$) can be calculated by effective medium theory (EMT), which models the porous material as a typical two-phase structure. In Equation (4), EMT assumes a random distribution of two phases, where (1) refers to parent material (copper) and (2) refers to pores. By using the Wiedemann–Franz law (Equation (5)), EMT could be reformulated to describe the relationship between σ and ε. In the case of porous copper, Koh calculated the material-dependent numbers L and b to be 2.307 × 10^−8^ $W\Omega /K^{2}$ and 18.6 W/m.K, respectively [34]. At ε = 0.06, which is approximately the porosity % of our printed coils, the predicted $\sigma \left(IACS\right)$ by means of the aforementioned models (Equations (3a), (3b), and (4)) is 87%, 93.3%, and 89%, respectively. Thus, the obtained σ of the printed coils shows 10% less than the average of those estimated values. There are several possible reasons behind this contradiction: (1) disparity of pores distribution and shape, (2) none of these models account for the presence and size of grain boundaries. The grain boundaries of components produced using laser additive manufacturing are superior to those of parts treated using powder metallurgy or other conventional methods. Additionally, the presence of impurities could be responsible for the decline in σ [3]. These models, in contrast, presupposed high-purity copper.

The AC resistance (R_ac_) spectrum is plotted in Figure 14c for the Cu-printed and solid coils with respect to a frequency range from 20 up to 10^5^ Hz. It can be seen in all curves that the resistance is continuously rising along the frequency range. Although square wires have a larger cross-sectional area than round wires, square wires’ resistance increases dramatically at high frequencies. It is worth noting that the difference in R_ac_ between printed and solid coils in both square and round cases tends to be constant at low frequencies, as seen in the magnified graph in Figure 16d, and then continuously diverges by increasing the frequency.

The tendency of AC to concentrate near the conductor’s outer rim is known by the “skin effect” rather than being distributed evenly throughout the whole cross section of the conductor. At high frequency, the current density is maximum (I_o_) at the circumference and reduces exponentially toward the center. As a result, it increases the wire resistance by reducing the effective area, as illustrated in Figure 16e. “Skin depth” is defined by the distance from the outer surface to the point at which the current intensity reaches 37% of I_o_ (Equation (6)) [36]. For example, the skin depth of LPBF copper wire is calculated to be 0.5 mm at 18 KHz. Unfortunately, as observed in Figure 15, the porosity intensity gradient from the center to the edge of the LPBF parts is positive, which is also the case for the current density distribution at high frequency. At this moment, the region with high porosity (high resistance) will accommodate the vast majority of the alternating current. Thus, the R_ac_ curve of the LPBF coil is steeper than that of solid wire. In addition, surface roughness negatively impacts conductor resistance, particularly when the skin depth becomes sufficiently shallow and comparable with the roughness scale [37]. It is attributed to the increase in charge carrier path length. Therefore, surface roughness possibly contributes to the higher resistance of printed coil, because its surface quality is inferior to the smooth surface of solid wire. In this study, the skin depth at 100 KHz is 200 μm, much higher than the R_a_ measured for printed samples. Thus, it is believed that porosity distribution is dominant here. The conductor’s cross-section geometry and shape considerably affect the current density distribution. As depicted in Figure 16e, the skin effect reduces the effective area of the square conductor by inducing “current crowding” toward the corners [38]. Accordingly, the R_ac_ of the solid and printed square coil is relatively higher, as found in Figure 16c.
(6)$$\delta =\sqrt{\frac{2}{\omega \mu \sigma }}$$
where δ is the skin depth, ω is the angular frequency, μ is the absolute magnetic permeability of the conductor.

### 3.3. Effect of HT

HT of the contoured Cu-LPBF samples led to further improvement in surface roughness. The Ra was reduced from 5.3 to 3.9 μm due to more fusion of previously adhered particles, as manifested by the SEM micrograph of the side surface in Figure 17. In comparison, the optimum R_a_ of the side surface of as-built samples obtained by employing a high-precision LPBF machine is 6.8 μm [17]. The applied HT effectively sintered the as-built components, resulting in the fusion of both the unmelted powders and the adjacent tracks [31]. This sintering process resulted in the development of neck formation between adhered particles and the side surface. It is reported that HT can reduce the surface roughness of AlSi10Mg parts manufactured by LPBF by 17% [39]. When heat is applied to powder particles near melting temperature, the particle shell starts to melt, but the core stays unaffected. It is known as “liquid phase sintering”. Melting and fusion of the exterior layer of the particle continue as long as the heat is applied near the melting point [40].

Figure 18 compares the σ of Cu processed by different powder bed AM techniques, including the results obtained in this study, with the min and max σ calculated by conductivity–porosity models (Equations (3)–(5)). It is noted that the σ of the part manufactured by LPBF in (d), (f), and the current work (b) approach (equal to or barely less than) the estimated values by σ−ε models at the corresponding RD, regardless of being in as-built or HTed condition. However, there are two exceptions to this trend, where the σ is either significantly low in (a) or exceeds both $\sigma_{min}$ and $\sigma_{max}$, as shown in (e). At relatively low laser power, incomplete fusion between tracks and deposited layers results in unmelted powder particles trapped inside the pores [31]. Therefore, due to multiple interfaces of powder particles and the pore’s interior, the electrical resistivity is further increased if compared with unfilled pores with the same empty volume. Hence, the shape, structure, and distribution of pores affect the σ, not just ε% [33]. This observation can also justify the noticeable rise in conductivity after HT, in which the number of unfused powder particles was decreased by partially melting and bonding them to the parent part [31]. On the other hand, electron beam-powder bed fusion (EB-PBF) is superior in the field of Cu additive manufacturing. A high-purity fully dense part with a long columnar grain structure accounts for the remarkable performance of EB-PBF [41], even though the conductivity values may be overestimated [3]. Employing high-precision (hp)-LPBF with 25 μm beam size results in a tiny grain size in the range of 5–7 μm and consequently higher resistivity. It occurs due to a smaller molten pool and a faster cooling rate, leading to fine grain size and high dislocation density [17]. HT at 1000 °C, higher than the recrystallization temperature, enhances conductivity by restoring grain boundaries and dislocation density, bringing it closer to $\sigma_{min}$, as in case (f) and the current work. Nevertheless, as reported in [3], the electrical conductivity without HT could reach the value of $\sigma_{min}$ (Figure 18d). In binder-jetting AM, the remaining carbon after binder burn-off contributes to further conductivity loss [33]. It is reasonable to conclude that the conductivity–porosity relationship is not always linear in Cu processed with AM and depends on multiple factors.

Cu-LPBF samples were cut, polished, and etched to study the microstructure evolution and justify the improvement in conductivity after HT, as shown in Figure 19. The average grain size was calculated according to ASTM E112-13 by substituting into Equation (7) [42]. The average grain size before and after HT was found to be 16.5 μm and 46.3 μm, respectively.
(7)$$l_{gb}=\sqrt{\frac{A}{n_{enc}+0.5n_{int}\pm 1}}$$
where *A* is the area of a rectangular portion taken from the microscope image, $n_{enc}$ and $n_{int}$ are the number of enclosed and intercepted grains with the perimeter.

This section aims to determine whether the change in conductivity and grain size development are correlated. The conductivity of the fully dense copper matrix processed with LPBF can be theoretically calculated by rearranging Equation (6) to take the form of $k_{1}=f(K_{eff},k_{2},v_{2})$. In this instant, $K_{eff}$ is the calculated conductivity of the printed coil (from measured resistance) at the corresponding porosity $v_{2}$. $k_{1}$ is now accounting for the effects of grain boundaries, impurities, and any other factor except porosity will be noted as $k_{npd}$ “non-porosity defect”. It is equal to 365 W/m.k and 388 W/m.k for as-built and HTed coils, respectively. According to Equation (8), as reported in [33], the thermal resistivity of a polycrystalline matrix is the summation of the grain boundaries and single-crystal resistance. Therefore, the ratio of grains number per unit length before and after HT can be related to the change in conductivity by Equation (9). This relation is formulated based on several assumptions as follows: (1) no significant alteration in the constituent elements of the grain interface before and after HT, (2) the conductivity–porosity relation of Cu-LPBF follows the EMT model, and (3) the influence of other defects compared with grain boundaries could be negligible. Substituting into Equation (9) indicates a good correlation between the obtained conductivity after HT and grain growth, where k_single_ is the standard conductivity of copper (398 W/m.k) [43].
(8)$$k_{poly}={\left(\frac{1}{k_{single}}+nR_{th}\right)}^{-1}$$
where *k_poly_* and *k_single_* are the thermal conductivity of polycrystalline matrix and single-crystal structures, respectively. *R_th_* is the interface’s resistance. *n* is the number of interfaces per unit length (1/$l_{gb}$).
(9)$$\frac{n_{HT}}{n_{As-built}}\approx \frac{{\left[\frac{k_{single}}{k_{npd}}-1\right]}_{HT}}{{\left[\frac{k_{single}}{k_{npd}}-1\right]}_{As-built}}$$

## 4. Conclusions and Future Work

The present work aims to improve the power density of the Cu-LPBF coil by enhancing the SFF and the electrical conductivity. The minimum spacing between the TF was improved to be around 80 μm after applying sample contouring and the optimum beam offset of 110 μm. The SFF was calculated to be 79% and 63% for square and round wire, respectively. The σ of as-built LPBFed Cu winding was 20% less compared with the standard conductivity of Cu. As a side benefit, surface roughness significantly improved by applying both sample contouring and HT, where R_a_ reduced from 9.5 to 4.9 μm. The difference in R_ac_ between printed and solid coils in both square and round cases was observed to be constant at low frequencies and then continuously diverged by increasing the frequency. HT increased the conductivity by 7% (IACS). A good correlation between the obtained conductivity and grain growth after HT was demonstrated.

Further investigation of the electrical performance needs to be carried out on the printed coils under high-power conditions. This involves applying high-current circuits to simulate the actual operating conditions of electrical motors. Exploring different types of insulation materials is a valuable future direction. Various insulating materials, such as polymers, ceramics, or composite coatings, can be applied to improve electrical insulation and thermal management.

## Figures and Tables

**Figure 1 materials-16-04694-f001:**
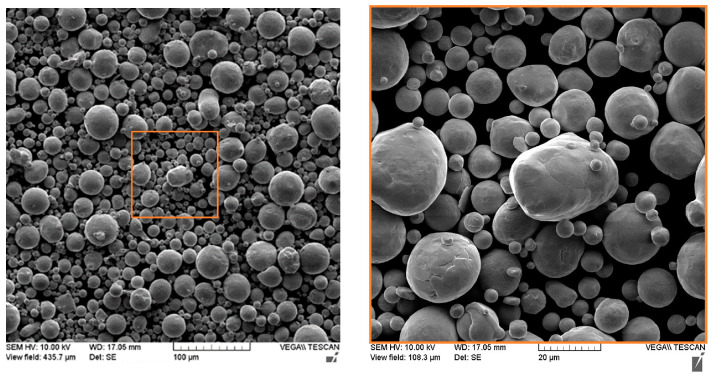
Scanning electron microscope image of copper powder.

**Figure 2 materials-16-04694-f002:**
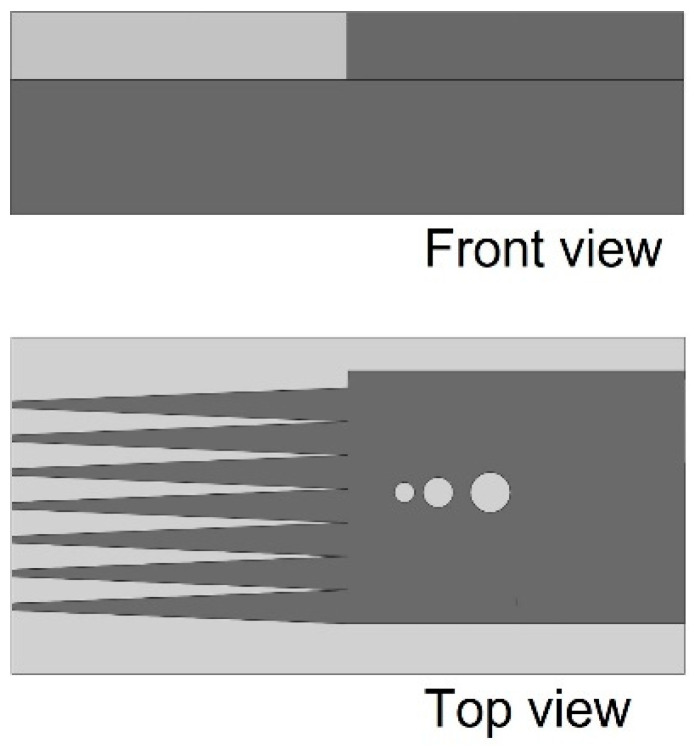
CAD designs of the coupon used for geometry assessment.

**Figure 3 materials-16-04694-f003:**
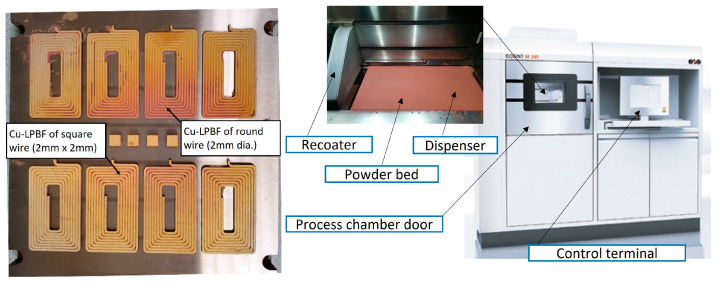
As-built Cu samples before separation from the steel build plate (**left**) and EOS M280 (**right**).

**Figure 4 materials-16-04694-f004:**
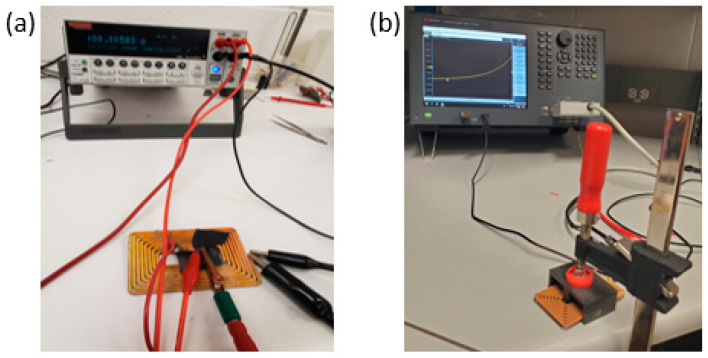
Resistance measurement setup for copper coils: (**a**) DC resistance, (**b**) AC resistance using the impedance analyzer.

**Figure 5 materials-16-04694-f005:**
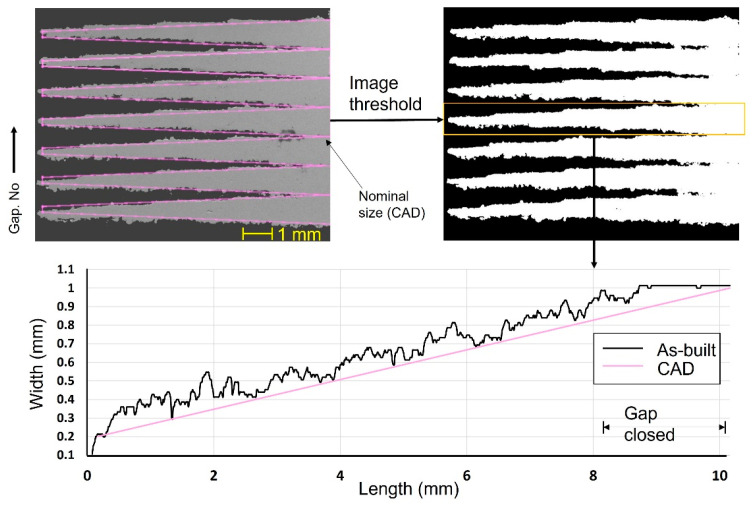
Width variation of the TF before contouring and beam offset adjustment.

**Figure 6 materials-16-04694-f006:**
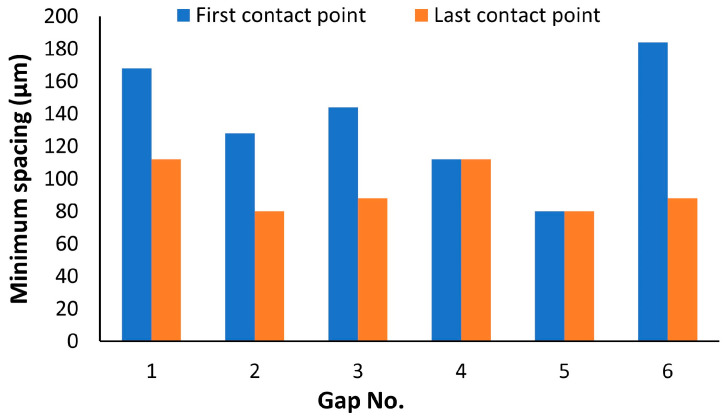
Minimum spacing based on gap closing at the first and last contact points.

**Figure 7 materials-16-04694-f007:**
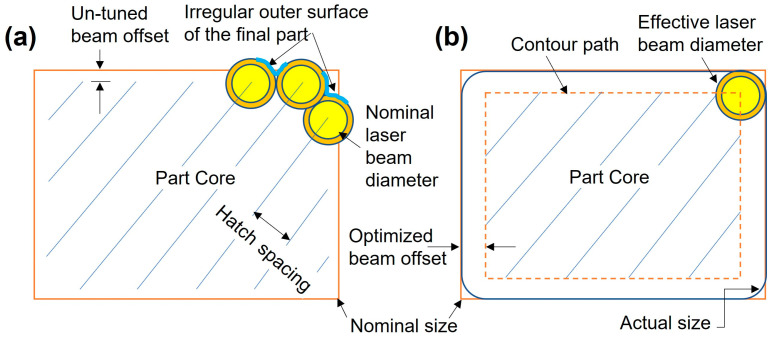
Schematic of LPBF part size in case of (**a**) untuned beam offset (machine default) and without contouring, (**b**) with contouring and optimized beam offset.

**Figure 8 materials-16-04694-f008:**
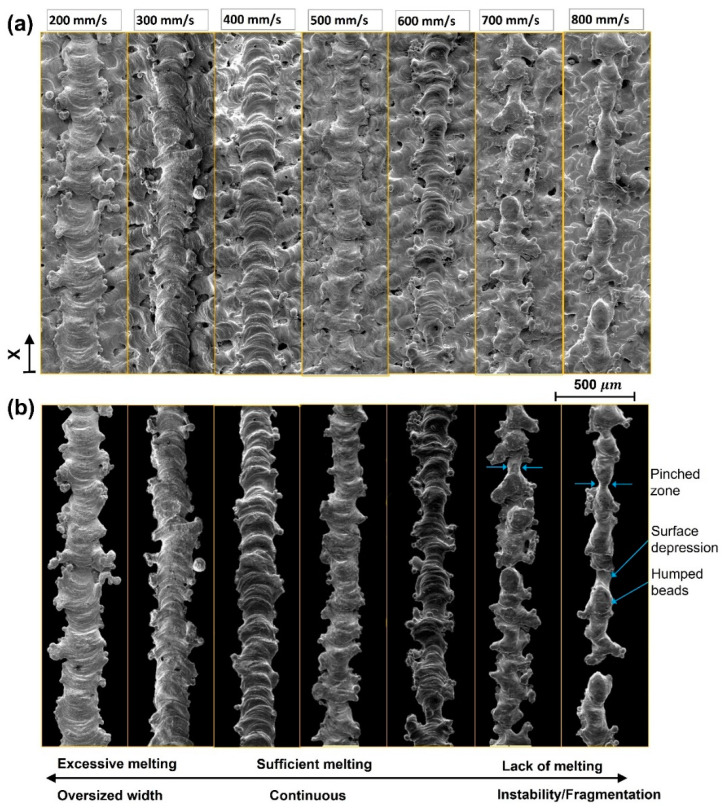
Single tracks of Cu-LPBF show the evolution of their morphology with respect to scanning speed: (**a**) SEM images of scan lines deposited on top of printed coupons, (**b**) scan lines with a black background.

**Figure 9 materials-16-04694-f009:**
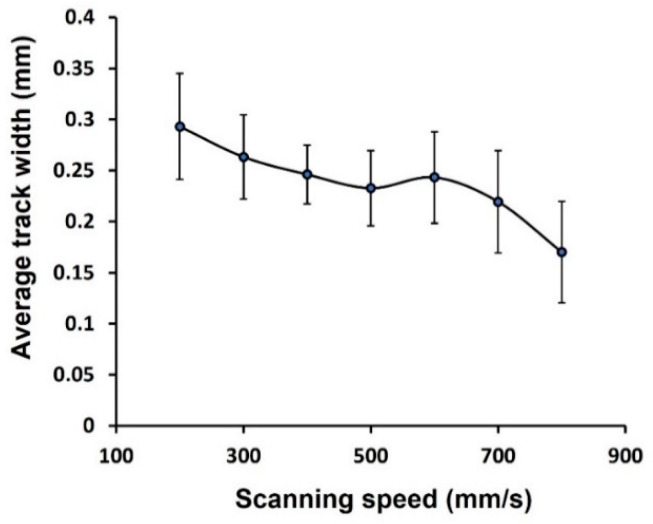
Single-track average width as a function of scanning speed.

**Figure 10 materials-16-04694-f010:**
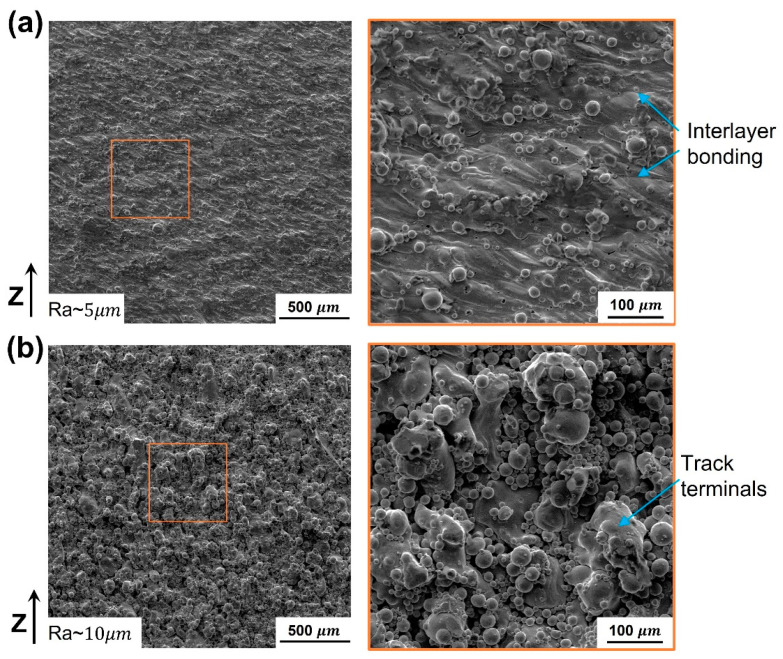
SEM micrograph of the side surface Cu-LPBF samples: (**a**) with contouring, (**b**) without contouring.

**Figure 11 materials-16-04694-f011:**
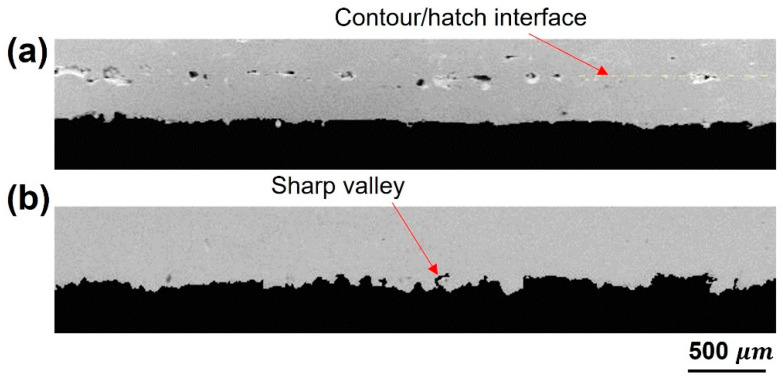
SEM of Cu-LPBF samples edge: (**a**) with contouring, (**b**) without contouring.

**Figure 12 materials-16-04694-f012:**
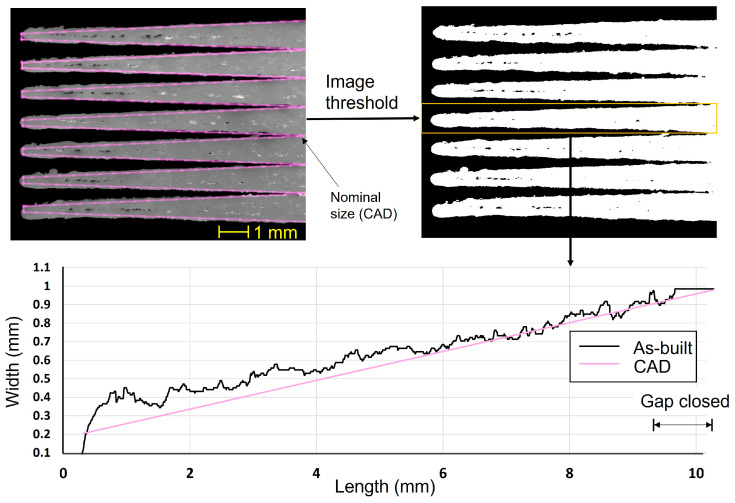
Width variation of the TF after contouring and beam offset adjustment.

**Figure 13 materials-16-04694-f013:**
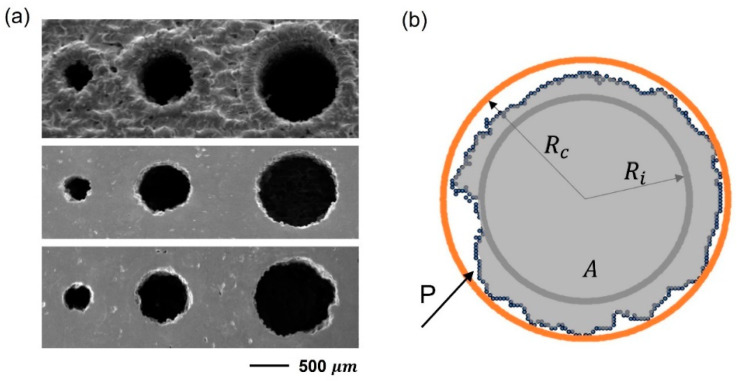
Roundness of microholes fabricated on Cu-LPBF coupon: (**a**) SEM of s-built top surface and polished surfaces at two positions along the building direction (top to bottom), (**b**) illustration of the parameters used in the two mentioned methods of roundness calculation.

**Figure 14 materials-16-04694-f014:**
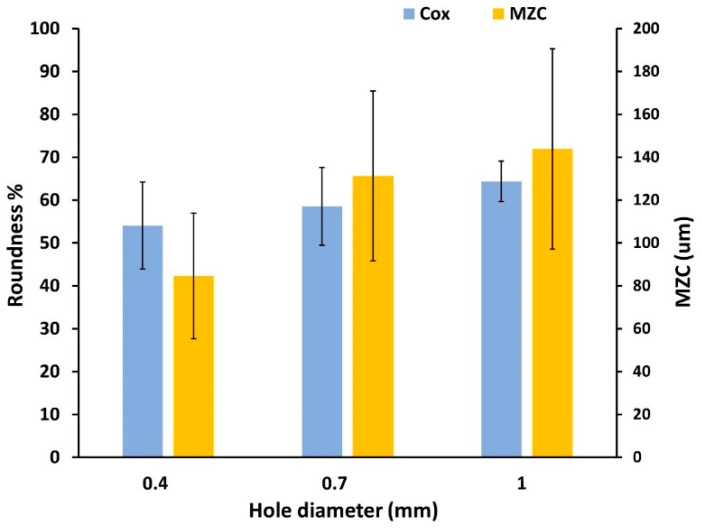
Roughness error of microholes fabricated on Cu-LPBF coupon using Cox’s and the MZC methods.

**Figure 15 materials-16-04694-f015:**
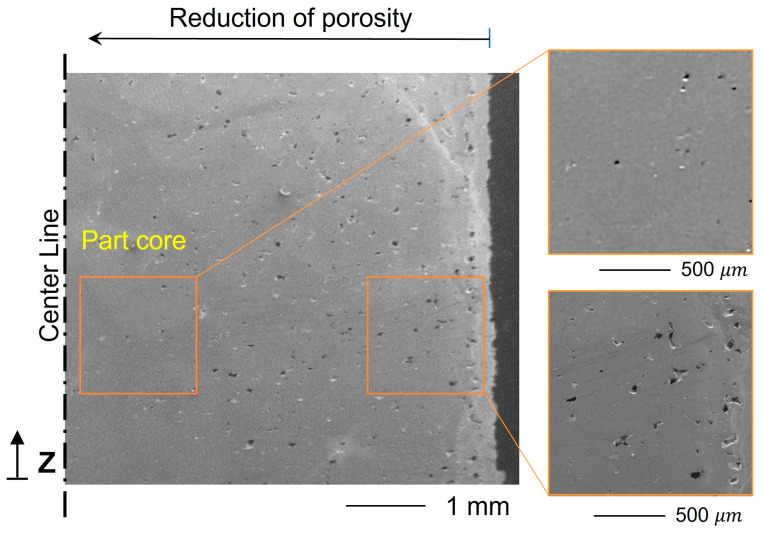
SEM of Cu-LPBF cross section shows the distribution of porosity in the transverse direction.

**Figure 16 materials-16-04694-f016:**
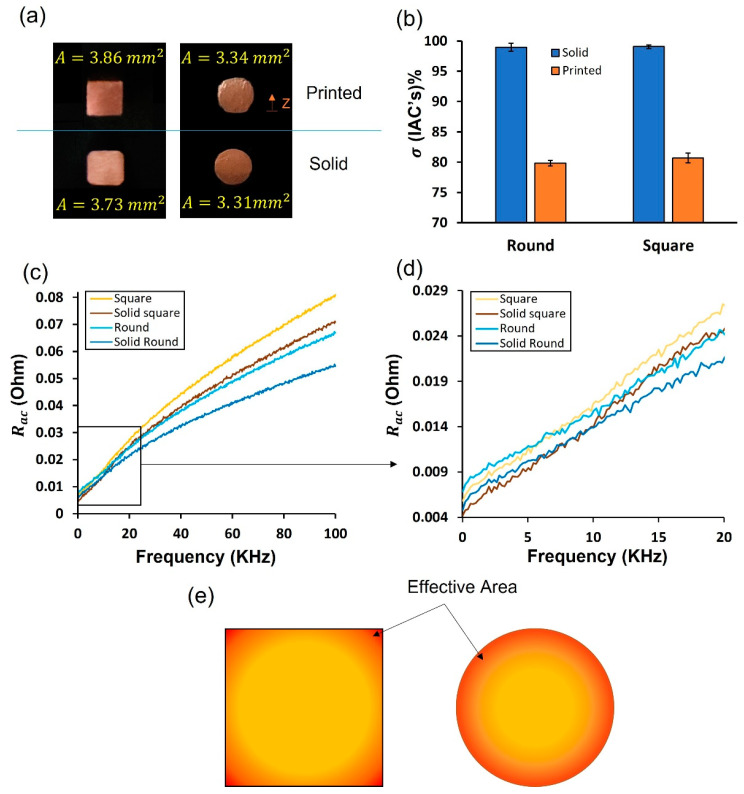
DC and AC resistance of solid and printed coils. (**a**) The actual cross-sectional areas of the used wires. (**b**) Electrical conductivity of LPBFed round and square coils compared with solid. (**c**) The AC resistance of the four coils. (**d**) Magnified spectrum at low frequencies. (**e**) Illustration of the effective area at high frequencies known as “the skin effect”, which coincides with the high-intensity region of LPBF part defects at the borders.

**Figure 17 materials-16-04694-f017:**
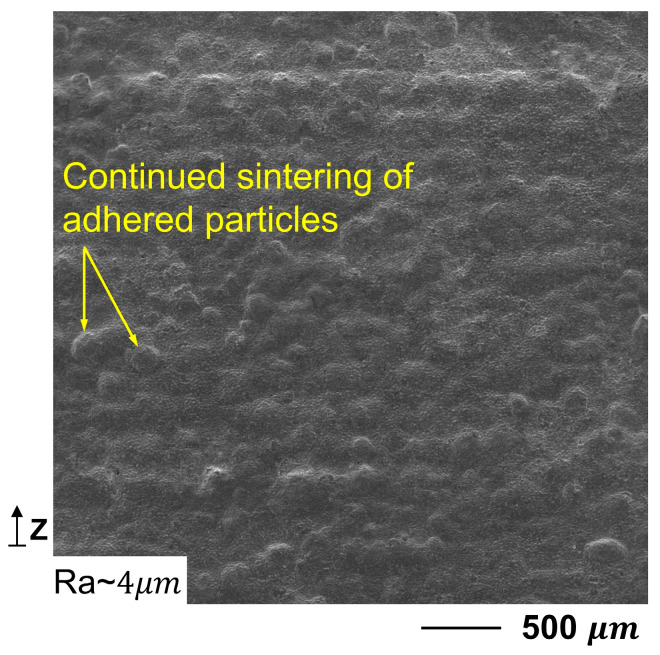
SEM micrograph of the side surface after HT shows improvement of Cu-LPBF’s surface roughness.

**Figure 18 materials-16-04694-f018:**
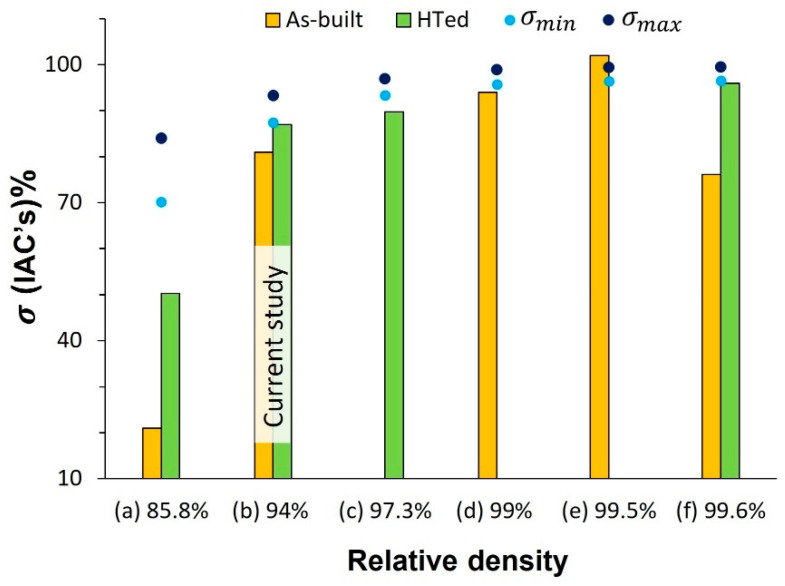
Comparison of the electrical conductivity of Cu processed by different powder bed AM methods: (**a**) medium-power LPBF [31], (**b**) current study, (**c**) binder-jetting AM [33], (**d**) high-power LPBF [3], (**e**) EB-PBF [41], and (**f**) high-precision LPBF [17] with min and max calculated σ by conductivity–porosity models.

**Figure 19 materials-16-04694-f019:**
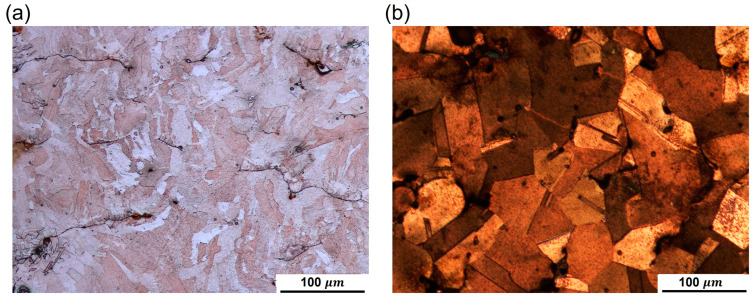
Microstructure before (**a**) and after (**b**) HT.

**Table 1 materials-16-04694-t001:** Beam offset and scanning speed assigned for contouring and core Cu-LPBF.

	Scanning Speed (mm/s)	Scanning Orientation	Beam Offset (mm)
Contour	400	NA	0.11
Core	500	67°	0.15

**Table 2 materials-16-04694-t002:** Conductivity–porosity models.

Ref.		Conductivity and Porosity Relationship	
[32]	(a) ω=2.1	$\sigma =\sigma_{o}(1-\omega \varepsilon )$	(3)
[33]	(b) ω=1.123		
[35]	$K_{eff}=0.25[\left(3v_{2}-1\right)k_{2}+\left(2-3v_{2}\right)k_{1}+\sqrt{{\left\{\left(3v_{2}-1\right)k_{2}+\left(2-3v_{2}\right)k_{1}\right\}}^{2}+8k_{1}k_{2}}]$ where v is the volume fraction		(4)
[34]	k=LTσ+b (Wiedemann–Franz law) Where *L* is the Lorenz number, *T* is the absolute temperature, and *b* is the material-dependent lattice (phonon) contributions to thermal conductivity		(5)

## Data Availability

Data is not available for publication.

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
