# Peer review of "On the Fabrication of High-Performance Additively Manufactured Copper Winding Using Laser Powder Bed Fusion"

_materials, 2023, doi:10.3390/ma16134694_

Round 1

Reviewer 1 Report

This paper presents the fabrication of high-performance additively manufactured copper winding using LPBF. Some issues need to be addressed as following:

1.     What is the purpose of Fig.1? Maybe the distribution of dimension of the powder is better to illustrate the effects of the powder dimension.

2.     The cross-section in Fig.3 is not very clear.

3.     Does the laser power change in this paper? In this paper, the effects of the scanning speed are discussed, is the interaction between laser power and scanning speed considered in this paper?

4.     Please explain how to improve the surface roughness by the heat treatment? And what is the relationship between the roughness and the conductivity?

Minor editing of English language required.

Author Response

Dear reviewer

I hope this message finds you well. I am writing to express my sincere appreciation for the time and effort you dedicated to reviewing my paper titled [On The Fabrication Of High-Performance Additively Manufactured Copper Winding Using Laser Powder Bed Fusion]. Your thoughtful and thorough review has significantly contributed to the improvement of my work, and I am truly grateful for your invaluable feedback.

Here are the answers to the mentioned points:

  1.     What is the purpose of Fig.1? Maybe the distribution of dimensions of the powder is better to illustrate the effects of the powder dimension.

The purpose of Figure 1 is to show the particle morphology (irregularities) and the adhesion of small particles . I added a high mag image for better presentation. Thanks for the note

  1. The cross-section in Fig.3 is not very clear.

yes, using "cross-section" word is confusing for readers, therefore i changed the label to square wire (2mmx2mm) and round wire (2mm dia.). also the cross-section of both wires were illustrated in fig. 16. Thanks for the note

  1. Does the laser power change in this paper? In this paper, the effects of the scanning speed are discussed, is the interaction between laser power and scanning speed considered in this paper?

Optimization of relative density and the impact of each parameter has been thoroughly investigated in our previous paper “Process–structure–property relationships of copper parts manufactured by laser powder bed fusion”. the previous study emphasizes the usage of the maximum available laser power on EOS M280 which is 370 W. so it is fixed in the current study. on the other hand, the effect of scanning speed on the morphology of single tracks was investigated aiming to improve the morphology of contour line ( used in sample contouring). Table 1 shows the assigned parameters used for contouring and core printing. in conclusion, the optimum parameters obtained in previous study still have been used here for printing the samples core , only the scanning speed of contour line was slightly changed. i mentioned this in the material and method section and table 1. Thanks for the note

  1. Please explain how to improve the surface roughness by the heat treatment? And what is the relationship between the roughness and the conductivity?

heat treatment near melting point for a prolonged time led to more fusion of adhered particles. Further details and discussion have been added. Please see section 3.3

there is no direct relationship between the surface roughness and conductivity. the goal is to improve the conductivity by heat treatment, but as a side benefit the surface roughness is improved as well.

i rewrote some parts of the abstract, introduction, and conclusion to state that the main objective is to improve the electrical power density of printed coil by increasing the slot filling factor (SFF) and the electrical conductivity. noted that in the electrical engineering field, Power density and current density usually represent the amount of electrical current carried by the coil per unit area.

The SFF can be optimized by reducing the spacing between coil turns. In this regards, the DA and surface quality need to be optimized first. on the other hand, the electrical conductivity can be improved by heat treatment (HT).

as a side benefit, surface roughness is noted to be improved by sample contouring and HT. so it is win-win situation.

At certain conditions, there is a relation between surface roughness and resistance. at ultra-high frequency application (Ghz), surface roughness is important to determine the resistance. but is not our case ( i mentioned this in the paper ), where the frequency is 100 KHz max (electrical motors are in the range of 1 KHz)

Thanks for the note

Reviewer 2 Report

This is a timely effort by authors on" On The Fabrication Of High-Performance Additively Manufactured Copper Winding Using Laser Powder Bed Fusion". However, there are few suggestions to improve this manuscript as this manuscript has many jargons. 

1. Novelty needs to be highlighted in a better way.

2. No.of references must be latest I.e. published in lastsr 5years as currently no. Of outdated references is more than 15.

3. Please show EOS M280 LPBF machine with proper captions.

4. Was the CAD design of Fig. 2 standardized?

5. Is it 2 mm or 2 mm^2 cross section in Fig. 3?

6.  Please mention the length, width, minimum spacing, and contouring for CAD design with reference to Fig. 5.

7.  What is h in Fig. 7?

8. In Fig. 16, a & b are mentioned wrongly! 

9. Figure 18 shows the literature based comparison of conductivities obtained via various AM techniques. Plz compare the roughness etc. Results with those from literature. 

10. Plz write the future work after conclusion section.

Minor but intensive spelling mistakes were observed which I dictates lack of essential proofreading. 

Author Response

Dear reviewer

I hope this message finds you well. I am writing to express my sincere appreciation for the time and effort you dedicated to reviewing my paper titled [On The Fabrication Of High-Performance Additively Manufactured Copper Winding Using Laser Powder Bed Fusion]. Your thoughtful and thorough review has significantly contributed to the improvement of my work, and I am truly grateful for your invaluable feedback.

Here are the answers to your suggestions:

  1. Novelty needs to be highlighted in a better way.
    i rewrote some parts of the abstract, introduction, and conclusion to point out the scope and contribution of this work. The gap in the literature is also highlighted. Thanks for the note.
  2. No.of references must be latest I.e. published in lastsr 5years as currently no. Of outdated references is more than 15.

I have revised the cited references. Some of them can be replaced or removed, as listed below

  • Cox, E. A method of assigning numerical and percentage values to the degree of roundness of sand grains. Palaeontol. 1927, 1, 179-183.
  • Jafferson, J.; Hariharan, P. Investigation of the quality of microholes machined by µEDM using image processing. Manuf. Processes 2013, 28, 1356-1360.
  • Sui, W.; Zhang, D. Four methods for roundness evaluation. Procedia 2012, 24, 2159-2164.
  • Özbilen, S. Satellite formation mechanism in gas atomised powders. Powder Metall. 1999, 42, 70-78.
  • Curran, B.; Ndip, I.; Guttowski, S.; Reichl, H. On the quantification and improvement of the models for surface roughness. In Proceedings of the Workshop on Signal Propagation on Interconnects (SPI), Strasbourg, France, May, 2009; pp. 1-4.
  • Tuncer, E.; Neikirk, D.P. Efficient calculation of surface impedance for rectangular conductors. Lett. 1993, 29, 2127-2128.

The other references describe important phenomena or possess unique properties which are highly relevant to the research topic and contribute significantly to the understanding of the subject matter. These papers have been widely cited and have become fundamental references in the field.

Thanks for the note

  1. Please show EOS M280 LPBF machine with proper captions.

Added, please see Fig. 3. Thanks for the note

4. Was the CAD design of Fig. 2 standardized?

It is a custom artifact to investigate specific dimensional accuracy of some features. There are too many proposed standards to investigate the dimensional accuracy in AM. As you can see in reference below , there are over 60 benchmark artifacts . Most of them are complex and have unwanted features. They studied different other properties such as flatness, concentricity, Parallelism .. etc

“A review on benchmark artifacts for evaluating the geometrical performance of additive manufacturing processes”

Thanks for the note

  1. Is it 2 mm or 2 mm^2 cross section in Fig. 3?

Revised, please see Fig. 3. Thanks for the note

6.  Please mention the length, width, minimum spacing, and contouring for CAD design with reference to Fig. 5.
The length of the trapezoidal feature (TF) is shown in the plot in Fig 5 which is 10 mm (pink line). The width of TF and the geometry of the sunken triangle (the tip of the triangle is the minimum spacing which is zero)  between TF are described in section 3.1.1. I just didn’t want to put many labels on the figure. No contouring is made in Fig. 5. The feature in Fig. 12 is contoured.

Thanks for the note

  1. What is h in Fig. 7?
    Replaced with hatch spacing, please see Fig. 7. Thanks for the note
  2. In Fig. 16, a & b are mentioned wrongly! 
    Fixed, Thanks for the note
  3. Figure 18 shows the literature based comparison of conductivities obtained via various AM techniques. Plz compare the roughness etc. Results with those from literature.

A comparison is made in section 3.3 . there is very limited data on the side surface roughness of copper part made by LPBF (one ref is found). Regarding the effect of heat treatment, there is non so I used ALSi10Mg ref. Thanks for the note

  1. Plz write the future work after conclusion section.

The future work has been added. Thanks for the note

Reviewer 3 Report

Moderate editing of English language is required

Author Response

Dear reviewer

I hope this message finds you well. I am writing to express my sincere appreciation for the time and effort you dedicated to reviewing my paper titled [On The Fabrication Of High-Performance Additively Manufactured Copper Winding Using Laser Powder Bed Fusion]. Your thoughtful and thorough review has significantly contributed to the improvement of my work, and I am truly grateful for your invaluable feedback.

Round 2

Reviewer 2 Report

The authors have now improved the manuscript and this manuscript can now be accepted in current form.

Reviewer 3 Report

The authors have properly replied to all my suggestions. Therefore, the paper can now be accepted in its current status.

Minor editing of English language required